# PolyVoice: Language Models for Speech to Speech Translation

**Qianqian Dong**[*], **Zhiying Huang**[*], **Qiao Tian, Chen Xu, Tom Ko**[†],
**Yunlong Zhao, Siyuan Feng, Tang Li, Kexin Wang, Xuxin Cheng, Fengpeng Yue,**
**Ye Bai, Xi Chen, Lu Lu, Zejun Ma, Yuping Wang, Mingxuan Wang, Yuxuan Wang**
ByteDance
{dongqianqian, huangzhiying.92, tom.ko}@bytedance.com

## Abstract

With the huge success of GPT models in natural language processing, there is a growing interest in applying language modeling approaches to speech tasks. Currently, the dominant architecture in speech-to-speech translation (S2ST) remains the encoder-decoder paradigm, creating a need to investigate the impact of language modeling approaches in this area. In this study, we introduce PolyVoice, a language model-based framework designed for S2ST systems. Our framework comprises three decoder-only language models: a translation language model, a duration language model, and a speech synthesis language model. These language models employ different types of prompts to extract learned information effectively. By utilizing unsupervised semantic units, our framework can transfer semantic information across these models, making it applicable even to unwritten languages. We evaluate our system on Chinese → English and English → Spanish language pairs. Experimental results demonstrate that PolyVoice outperforms the state-of-the-art encoder-decoder model, producing voice-cloned speech with high translation and audio quality. Speech samples are available at https://polyvoice.github.io.

## 1 Introduction

Speech-to-speech translation (S2ST) is a challenging task as it encounters all the difficulties of automatic speech recognition (ASR), machine translation (MT) and text-to-speech (TTS) synthesis. The research in S2ST focuses on two approaches: cascade solutions (Lavie et al., 1997; Baldridge, 2004; Nakamura et al., 2006) and direct solutions (Jia et al., 2019; 2022a). The advantage of cascade solutions lies in the convenience of improving the performance of individual modules, while direct solutions excel in lower latency and simpler model architectures. As for the direct S2ST solutions, it used to involve direct output of mel-spectrogram features (Dong et al., 2022) in the early stages, but recently, there has been a growing interest in predicting discrete units (Lee et al., 2022a). The unit-based approach has become more popular due to several reasons: (1) It eases the modeling difficulty of emitting spectrogram. (2) Units can be generated through unsupervised methods and can cover unwritten languages. (3) It allows a connection with other token-based NLP models.

Recently, language modeling (LM) approaches have made a lot of breakthroughs in natural language processing (NLP) (Zhao et al., 2023). The success of GPT models (Brown et al., 2020; Ouyang et al., 2022) is leading the community to a new era. Currently, the encoder-decoder models remain widely used in speech tasks, and the exploration of using LM approaches is still in its early stages. In fact, there have been attempts on ASR (Fathullah et al., 2023) and TTS (Wang et al., 2023), indicating that this direction is promising. Thus, we are motivated to investigate the performance of language modeling approach in S2ST. In this paper, we propose a semantic unit-based framework for S2ST system. Our framework (Fig. 1) consists of three LMs: a translation LM, a duration LM and a speech synthesis LM. The translation LM processes the semantic units of the source language and translates them into semantic units of the target language. The duration LM predicts the duration

---

[*]Equal contribution.
[†]Corresponding author.

information of the target semantic units and extends the unit sequence. The speech synthesis LM predicts the target acoustic units which are then converted into a waveform by a unit vocoder.

We employ various prompt types to extract the acquired knowledge from the language models utilized in our approach. Specifically, we concatenate the source and target semantic units, along with the source acoustic units, forming a comprehensive prompt for the speech synthesis language model. This enables the speech synthesis language model to grasp the acoustic characteristics of the source speaker and generate voice-cloned acoustic units accordingly. Importantly, both the semantic and acoustic units mentioned above are generated through unsupervised methods, making our framework applicable to unwritten languages.

We evaluate our system on Chinese → English and English → Spanish language pairs. Experimental results show that our system can generate voice-cloned speech with high translation quality and audio quality. We summarise our contribution as follows:

- We propose using a series of decoder-only language models to fulfill the S2ST task, whereas encoder-decoder models are the dominant structure in previous works.

- Unsupervised speech units are used in the framework and thus PolyVoice can cover both written and unwritten languages.

The rest of this paper is organized as follows. Section 2 introduces related work. Details of our method are described in Section 3. Section 4 introduces our experimental setup and main results. Section 5 presents our ablation study. Finally, we conclude our work in the last section.

## 2 RELATED WORK

### 2.1 SPEECH TOKENIZATION

There are two kinds of discretized speech units used in our work: semantic and acoustic units. Semantic units are usually derived from representations produced by speech encoder models like HuBERT (Hsu et al., 2021), mHuBERT (Lee et al., 2022c) or w2v-BERT (Chung et al., 2021). They capture the phonetics and semantic content in speech. Although the making of these units is originally developed to be used as target for training the speech encoder, recently there are attempts to directly use these units as input/output for semantic tasks (Kharitonov et al., 2021; Lakhotia et al., 2021; Meng et al., 2023; Zhang et al., 2023a). Acoustic units can also be referred to as codec units. They are originally developed to transmit high-quality speech signals under limited bandwidth. AudioLM (Borsos et al., 2023) is a pioneer work in using language models (LM) for audio generation. They make use of both kinds of units and build several LMs with different resolutions. VALL-E (Wang et al., 2023) further extends the AudioLM framework and applies it in TTS. They successfully demonstrate that the in-context learning capabilities of LM can be similarly replicated in the context of phoneme and codec units. In contrast to phoneme units which have to involve a supervised training process, both semantic and acoustic units can be generated through unsupervised methods.

### 2.2 TTS

Zero-shot cross-lingual TTS (Jia et al., 2018; Cooper et al., 2020) aims to build a system that can synthesize speech with user's voice and in a specific language that the user doesn't speak. Early attempts include speaker adaptation (Chen et al., 2019) and speaker embedding (Liu & Mak, 2019) approaches. LM-based TTS has been recently proposed and demonstrated promising results. VALL-E (Wang et al., 2023) introduces an LM-based approach that leverages in-context learning for zero-shot TTS. Their approach utilizes phoneme units and source acoustic units to prompt the LM in predicting the target acoustic units. The most relevant related work to ours is VALL-E X (Zhang et al., 2023b), which extends the VALL-E framework to tackle the cross-lingual problem. In their work, they concatenate the source and target phoneme units, along with the source acoustic units, to create a prompt for the LM.

## 2.3 S2ST

Speech-to-speech translation (Lavie et al., 1997; Baldridge, 2004; Nakamura et al., 2006) aims to develop models capable of generating target language speech from source language speech. A vanilla system traditionally employs a pipeline (Nakamura et al., 2006) that sequentially processes the input through automatic speech recognition (ASR) models, machine translation (MT) models, and text-to-speech synthesis (TTS) models. Recently, end-to-end paradigms (Jia et al., 2019; 2022a) have gained popularity in the field of S2ST, as they allow for a single model to perform one or more of the aforementioned tasks, which consequently reduces error propagation and latency. Among the various techniques, auxiliary supervision based on textual data has been particularly effective during training (Jia et al., 2019; Kano et al., 2021). However, this approach is not feasible when dealing with unwritten languages. To address this challenge, discrete units (Hsu et al., 2021) extracted from the speech are used to replace the target text, and then can be synthesized into the speech (Tjandra et al., 2019; Zhang et al., 2021; Lee et al., 2022a). Large scale studies have shown the powerful performance in various speech processing tasks (Nguyen et al., 2022).

Current research in speech-to-speech translation primarily emphasizes translation quality, with notable improvements observed in automatic evaluation metrics (like BLEU) or human evaluation of naturalness. However, there remain two persistent challenges in developing practical systems. First, these systems are predominantly developed and evaluated on small-scale benchmarks, while real-world scenarios often involve large quantities of labeled data, including ASR, MT, and S2T data (Agrawal et al., 2023). Even for low-resource or unwritten languages, leveraging unlabeled speech or text can provide valuable information (Lee et al., 2022a). Therefore, developing a unified model that jointly utilizes various data types is a critical research goal yet to be achieved. Second, while not a strict requirement, preserving the source speaker's style during translation is an important aspect of improving user experience (Zhang et al., 2023b; Song et al., 2023). However, capturing the unique characteristics of individual speakers is a challenging task. Current approaches, such as speaker embeddings (Jia et al., 2019) and multi-speaker TTS systems (Jia et al., 2018), have made some progress in this direction, but they are still far from practical requirements.

Taking a broad perspective, our work aligns with the framework presented in Lee et al. (2022a), where the source speech undergoes translation into discrete units before synthesizing it into the target language's speech. However, what distinguishes our work is the utilization of decoder-only language models to enhance the performance of each module. By leveraging diverse data sources within a language model-based framework, our proposed method effectively maintains the source speaker's style during synthesis, thereby demonstrating significant potential in practical systems.

## 3 METHOD

We present PolyVoice, an innovative language model-based framework for speech-to-speech translation, catering to both written and unwritten languages. Our proposed framework leverages discrete units, acquired through self-supervised training techniques such as HuBERT (Hsu et al., 2021), serving as an intermediary representation bridging the source and target speech modalities. PolyVoice provides a comprehensive framework consisting of two main components: a speech-to-unit translation (S2UT) front-end, facilitating the conversion of source language speech into target language units, and a unit-to-speech (U2S) back-end, skillfully synthesizing translated speech while preserving the personalized style of the source speaker. Figure 1 provides an illustrative overview of our approach.

### 3.1 SPEECH-TO-UNIT TRANSLATION (S2UT)

By employing discrete units obtained through self-supervised training, semantically irrelevant information from continuous speech representations is eliminated, facilitating effective training in an NLP paradigm. In this regard, the S2UT component leverages a language model to acquire the necessary cross-lingual generation capabilities based on the unit-based approach.

**Semantic unit extractor** S2UT initiates the processing of raw speech data by employing a sophisticated semantic unit extractor. Here we adopt HuBERT, which first encodes the speech by a stack of convolutions and Transformer layers to continuous representations at every 20-ms frame,

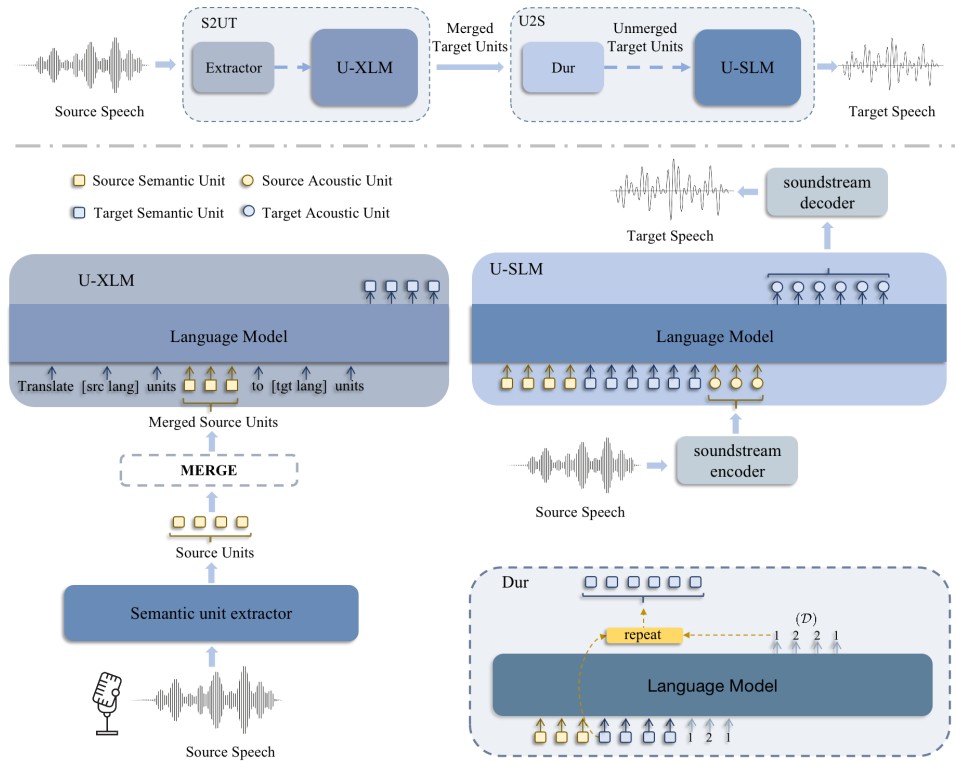

Figure 1: Overview of PolyVoice. The framework can be viewed as a concatenation of a speech-to-unit translation (S2UT) front-end and a unit-to-speech (U2S) back-end. There are three LMs in the framework: the unit-based cross-lingual LM (U-XLM), the duration LM and the unit-to-speech LM (U-SLM).

| | |
|---|---|
| **ASR: [lang]** | |
| **Data:** <unit, text> | |
| **Prompt1**: *Translate **[lang]** unit " **{unit}** " to **[lang]** text: " **{text}** "* | |
| **Prompt2**: *Translate **[lang]** text " **{text}** " to **[lang]** unit: " **{unit}** "* | |
| **MT: [src_lang] → [tgt_lang]** | |
| **Data:** <src_text, tgt_text> | |
| **Prompt**: *Translate **[src_lang]** text " **{src_text}** " to **[tgt_lang]** text: " **{tgt_text}** "* | |
| **S2ST: [src_lang] → [tgt_lang]** | |
| **Data:** <src_unit, tgt_unit, src_text, tgt_text> | |
| **Prompt1**: *Translate **[src_lang]** unit " **{src_unit}** " to **[tgt_lang]** unit: " **{tgt_unit}** "* | |
| **Prompt2**: *Translate **[src_lang]** unit " **{src_unit}** " to **[src_lang]** text: " **{src_text}** "* | |
| **Prompt3**: *Translate **[src_lang]** unit " **{src_unit}** " to **[tgt_lang]** text: " **{tgt_text}** "* | |
| **Prompt4**: *Translate **[src_lang]** text " **{src_text}** " to **[tgt_lang]** unit: " **{tgt_unit}** "* | |
| **Prompt5**: *Translate **[src_lang]** text " **{src_text}** " to **[tgt_lang]** text: " **{tgt_text}** "* | |
| **Prompt6**: *Translate **[tgt_lang]** text " **{tgt_text}** " to **[tgt_lang]** unit: " **{tgt_unit}** "* | |
| **Prompt7**: *Translate **[src_lang]** unit " **{src_unit}** " to **[src_lang]** text then **[tgt_lang]** text then **[tgt_lang]** unit: "**{src_text}** [sep] **{tgt_text}** [sep] **{tgt_unit}** "* | |
| *...* | |

Table 1: Data construction for the U-XLM model by various templates. These templates play a crucial role in generating multiple versions of training samples from diverse data resources such as ASR, MT, and S2ST, which are instrumental in facilitating cross-lingual unit generation.

and then utilizes k-means clustering to discretize the representation to a set of cluster indices $Z = z\_1, \cdots, z\_T$. $T$ is the number of frames and $z\_t \in [K]$, where $K$ is the number of cluster centroids. The discretized units are then merged by removing consecutive duplicated units.

**Unit-based cross-lingual language model (U-XLM)**  We denote the training sample consisting of units of speech in source language and target language as $<src\_unit, tgt\_unit>$. Within the encoder-decoder architecture, the encoder takes the source units as input, while the decoder generates predictions for the target units. To facilitate the generation of cross-lingual units, a straightforward approach involves utilizing simple prompts to construct training samples for natural language from unit pairs. For instance, one can create prompts like: "*Translate [src_lang] unit {src_unit} to [tgt_lang] unit: {tgt_unit}*". In addition to the direct transformation prompt, we can also instruct the model to generate intermediate steps similar to the cascaded systems in the *chain of thought prompting* (Wei et al., 2022) manner.

**Training**  To achieve competitive performance in training the U-XLM model, a large amount of data is crucial. However, obtaining supervised data, specifically cross-lingual unit pairs, is often limited in real-world scenarios. While auxiliary models can generate pseudo labels, such as synthesizing the target speech using the TTS model, direct training with supervised data is preferred.

To overcome the challenge of limited data availability, prior research has incorporated additional loss functions into the encoder-decoder architecture using multitask learning (Jia et al., 2022a; Lee et al., 2022a). In our work, we leverage the power of language modeling to address this issue in a more straightforward manner, allowing for the utilization of diverse data sources like automatic speech recognition (ASR) and machine translation (MT) data.

In Table 1, we demonstrate how we use different prompts to create training samples for different types of data sources. By employing parameter sharing and simplifying the design of auxiliary objectives, we train the model using these prompts. This approach also enables the direct utilization of unlabeled text and speech data. Consequently, the model implicitly enhances the alignment of the representation space between speech units and text.

## 3.2 UNIT-TO-SPEECH SYNTHESIS (U2S)

**Unit-to-speech language model (U-SLM)**  As illustrated in Figure 1, the U-SLM leverages the semantic units predicted by U-XLM and generates the codec units which incorporate the speaking style of source speaker. Similar to VALL-E X, U-SLM encompasses both an autoregressive model and a non-autoregressive model. However, instead of conventional phonemes, we employ discretized semantic units in our approach.

**SoundStream codec**  We employ SoundStream (Zeghidour et al., 2021), a neural audio codec, to generate embeddings of acoustic tokens. To ensure optimal performance, we re-implement the SoundStream with a hierarchy of 6 vector quantizers and a vocabulary of 1024 symbols. In our configuration, the acoustic tokens are produced at a rate of 80Hz for input waveforms sampled at 24 kHz, which results in a significant reduction in the sampling rate, specifically a reduction by a factor of 300 ($24000/80$). Once the U2S model predicts the acoustic tokens represented by the SoundStream codec, the decoder component of SoundStream reconstructs them back into the waveform.

**Duration model**  Through empirical observations, we have determined that the duration information of discretized units is crucial for ensuring stable and natural-sounding synthesized speech. In our approach, we employ an additional LM to predict the durations.

In Figure 1, we illustrate the process of incorporating duration prediction into our framework. The merged source semantic unit sequence, merged target semantic unit sequence, and the source duration value sequence ($\mathcal{D}$) are concatenated and provided as a prompt to the duration LM. Subsequently, the duration LM predicts the target duration value sequence, and each target semantic unit is repeated accordingly based on its predicted duration.

## 4 EXPERIMENTS

We conduct experiments utilizing a decoder-only model architecture following the standard GPT-2 (Radford et al., 2019). In Appendix A.2, we provide a comprehensive description of the model configurations employed in our work.

## 4.1 DATASETS AND PREPROCESSING

### 4.1.1 S2UT

**Semantic tokens**  U-XLM is trained by cross-lingual unit data, which is extracted from the audio by HuBERT (Hsu et al., 2021) models. For Chinese audio, we utilize an open-source model based on WenetSpeech Chinese speech[1]. For English and Spanish audio, we use an open-source multilingual model (English, Spanish and French)[2]. The cluster centroids of a k-mean algorithm for the two models are 500 and 1,000, respectively.

**Vocabulary**  To address the out-of-vocabulary problem and enable parameter sharing across languages, we utilize byte-level subword units[3] that decompose each character into byte-sized pieces and achieve a final vocabulary size of 56,407, including 1,500 cluster centroids (<zh-0>,...<zh-499> and <m-0>,...,<m-999>).

**Datasets**  Considering that the paired speech-to-speech (S2S) data is scarce, we synthesize the pseudo data from the ASR data utilizing in-house MT and TTS systems. In addition, various types of data resources provide better learning of the U-XLM model, like large-scale ASR and MT data. A more elaborate description of the used datasets can be found in Appendix Table 7.

The S2S data is sourced from WenetSpeech (Zhang et al., 2022) and GigaSpeech (Chen et al., 2021), respectively, marked as "GigaS2S" and "WenetS2S". WenetSpeech is a Chinese ASR dataset with over 10,000 hours of speech data collected from YouTube. And we utilize a subset of 10,000 hours of GigaSpeech (Chen et al., 2021), an English ASR dataset collected from audiobooks, podcasts, and YouTube.

Then we scale up the training data using specific prompts for various types of datasets. We utilize the LibriLight (Kahn et al., 2020) and the in-house ASR datasets. LibriLight is an unlabeled English speech dataset containing about 60,000 hours of speech. Since LibriLight has many long audios, we segment and recognize the audio based on the method of voice active detection (VAD) and in-house ASR system, generating the audio length ranging from 0.5 to 25s, and the average length is 7s. The in-house ASR dataset is a Chinese ASR dataset with 60,000 hours of speech. We also use the in-house Chinese-English MT dataset consisting of 44M sentence pairs.

### 4.1.2 U2S

The U-SLM is trained on the large open-source bilingual speech data, including WenetSpeech (Zhang et al., 2022) and LibriLight (Kahn et al., 2020). The Librilight is handled in the same way as U-XLM. WenetSpeech keeps the original data length unchanged. The duration of audio samples ranges from 0.5 to 20s, and the average duration is 2.5s. To further improve the synthesized quality, we use an additional 250-hour internal Chinese TTS data and 400-hour internal English TTS data.

## 4.2 EVALUATION

Our method is evaluated on two speech-to-speech benchmark datasets, EMIME (Wester & Liang, 2011) (Chinese → English) and CVSS (Jia et al., 2022b) (English → Spanish). Apart from the overall result, we report the separate performance on the S2UT front-end and U2S back-end. EMIME contains bilingual Chinese-English speech recorded by the same speakers. For CVSS, the translation speech is in voices automatically transferred from the corresponding source speech. To measure the performance of our system, we evaluate both the translation quality and the speech quality.

**Translation Quality**  Following the previous setups, we recognize the speech output by an in-house ASR system to compute BLEU scores (ASR-BLEU) for S2ST results using `sacrebleu`[4].

---

[1]https://github.com/TencentGameMate/chinese_speech_pretrain
[2]https://github.com/facebookresearch/fairseq/blob/main/examples/speech_to_speech/docs/textless_s2st_real_data.md
[3]https://github.com/huggingface/tokenizers
[4]https://github.com/mjpost/sacrebleu

| | ASV ↑ | | | ASR-BLEU ↑ | Naturalness ↑ |
|---|---|---|---|---|---|
| | tgt vs. src | hyp vs. src | hyp vs. tgt | | |
| Cascade (VALL-E X paper) | | 0.28 | 0.27 | 27.49 | 3.44 |
| + w/ oracle target text | | 0.28 | 0.29 | 80.30 | 3.43 |
| | 0.58 | | | | |
| VALL-E X (VALL-E X paper) | | 0.37 | 0.37 | 30.66 | 3.54 |
| + w/ oracle target text | | 0.39 | 0.38 | **86.78** | 3.54 |
| S2UT | | 0.06 | 0.08 | 29.30 | 3.35 |
| PolyVoice (S2UT + U2S) | 0.59 | 0.38 | 0.38 | 29.40 | 4.10 |
| + w/ chain of thought decoding | | 0.38 | 0.38 | 30.80 | 4.11 |
| + w/ oracle target semantic unit | | **0.42** | **0.48** | 76.10 | **3.92** |

Table 2: S2ST results on Chinese-English EMIME dataset. The **bold** and underlined numbers represent the best results of full-pipeline decoding. Only **bold** numbers signify the best synthesized results using the oracle target translation as input.

**Speech Quality**   The speech quality is evaluated by multiple metrics. The capability of voice clone is measured by the speaker similarity (ASV-Score), which is calculated by an ASV model[5] to determine whether the synthesized speech is from the same speaker as the ground-truth speech. The naturalness of the speech output is evaluated by the automatic metric using `NISQA`[6]. And the pronunciation accuracy is evaluated using WER scores (ASR-WER) with an ASR model based on `hubert-large`[7].

## 4.3 RESULTS AND ANALYSIS

### 4.3.1 S2ST RESULTS

Table 2 summarizes the overall performance of our method for S2ST. We conduct experiments on the EMIME dataset to enable direct comparisons with the most similar work VALL-E X. The cascade system treats S2ST as a pipeline of running an ASR model, an MT model, and a multi-speaker YourTTS model separately and sequentially. During the synthesis process, speaker information is integrated using speaker embeddings.

We first evaluate the capability to preserve the voice of the source speaker in the output speech, using the ASV score. We calculate speaker similarity between the source speech, target speech, and synthesized speech. We can run the U-XLM alone, where speech is synthesized by a Unit-based vocoder[8] (Lee et al., 2022c). Due to the lack of explicit modeling of speaker characteristics, it produces particularly low ASV scores. Both the VALL-E X and PolyVoice systems, which adopt in-context learning, show superior performance over the speaker embedding based method. Notably, our method demonstrates better voice cloning capabilities when ground-truth target information is available.

PolyVoice achieves a slightly enhanced translation quality (ASR-BLEU) but a remarkable improvement in speech quality (naturalness) compared with VALL-E X. When taking the ground-truth target information as input, PolyVoice is inferior to VALL-E X with a large gap of about 10 BLEU points, while the naturalness improves significantly. The semantic units are extracted from the speech by unsupervised learning, which inevitably introduces errors. Although units are considered "semantic" tokens, they still preserve some acoustic information. Therefore, unit-based modeling leads to better speech quality but worse translation quality. In contrast, phonemes obtained from the text ensure semantic correctness but lose the acoustic information. And future work can focus on enhancing the extraction of semantic information to improve translation quality.

---

[5]https://github.com/Sanyuan-Chen/UniSpeech/tree/t-schen/asv_eval/downstreams/speaker_verification#example-2
[6]https://github.com/gabrielmittag/NISQA
[7]https://huggingface.co/facebook/hubert-large-ls960-ft
[8]https://github.com/facebookresearch/fairseq/blob/main/examples/speech_to_speech/docs/textless_s2st_real_data.md

| CVSS | ASV ↑ | BLEU ↑ | Naturalness↑ |
|---|---|---|---|
| Ground-truth | 0.19 | 89.3 | 3.54 |
| PolyVoice | 0.34 | 18.3 | 3.60 |
| + w/ oracle target unit | 0.28 | 70.8 | 3.69 |

Table 3: Results on the English-Spanish CVSS dataset. We train the model with paired speech-to-speech datasets expanded from GigaSpeech without any text information. BLEU means ASR-BLEU, target unit means oracle Spanish unit.

| Arch | Training Data | ASR-BLEU |
|---|---|---|
| Encoder-Decoder | | 16.8 |
| + w/ U2S | GigaS2S | 18.7 |
| Decoder-only | WenetS2S | 20.7 (+3.9) |
| + w/ U2S | | 22.0 (+3.3) |

Table 4: Performance with different architectures on EMIME dataset.

| Task | S2ST (BLEU ↑) | ASR (CER ↓) | ST (BLEU ↑) | MT (BLEU ↑) | TTS (WER ↓) |
|---|---|---|---|---|---|
| S2S | 22.2 | - | - | - | - |
| + MTL | 29.4 | 4.46 | 30.8 | 33.81 | 6.99 |

Table 5: Performance with multi-task learning (MTL) on the EMIME dataset. Here are the explanations for each task. S2ST: Chinese speech to English speech; ASR: Chinese speech to Chinese text; ST: Chinese speech to English text; MT: Chinese text to English text; TTS: English text to English speech.

Interestingly, PolyVoice achieves better naturalness using the predicted units. We speculate that this is due to the language model's output having better fluency. U-XLM learns the speech distribution over the large scale of unit data, and tends to generate more natural sequences of units. However, this may interfere with the accuracy of the translation. We will explore this issue in the future.

### 4.3.2 UNWRITTEN LANGUAGE SCENARIO

We examine our proposed framework in the case where the source is a written language and the target is an unwritten language. In our setup, we train and evaluate an English→Spanish S2ST system without the use of any Spanish text transcript. Table 3 summarizes the results. The ASR-BLEU (18.3) indicates that the Spanish speech generated by our system is semantically understandable. This demonstrates the ability of our S2ST system for the unwritten languages.

## 5 ABLATION STUDY

### 5.1 DECODER-ONLY VS. ENCODER-DECODER

Empirical studies in the field of natural language processing have revealed that the full potential of the decoder-only approach can be realized through the use of large model sizes and expansive datasets. As pioneers in exploring the application of language models to S2ST, we present a fair comparison of the two architectures, *decoder-only* and *encoder-decoder*, in Table 4. Two frameworks are trained with the same training data and similar parameters, approximately 0.6B in size. Interestingly, the decoder-only model yields a remarkable improvement of 3.9 BLEU points over the encoder-decoder counterpart[9] (Lee et al., 2022b). When we synthesize the speech by U2S instead of vocoder, the performance gap is reduced, highlighting the robustness of our U2S back-end.

---

[9]The encoder-decoder architecture is experimented using the implementation: `https://github.com/facebookresearch/fairseq/blob/main/examples/speech_to_speech/docs/direct_s2st_discrete_units.md`.

| Methods | WER ↓ | ASV ↑ | Naturalness ↑ |
|---|---|---|---|
| VALL-E X (paper) | 4.07 | 0.36 | 3.54 |
| U2S | 6.40 | 0.38 | 3.98 |
| + w/o semantic2dur | 31.93 | 0.37 | 3.81 |
| + w/ mHuBERT_zh_en | 4.76 | 0.37 | 3.81 |

Table 6: Evaluation of the speech synthesizers.

## 5.2 MULTI-TASK LEARNING

As discussed in Section 3, the language modeling enables direct training over the diverse data sources utilizing specific prompts. In this way, we combine additional large-scale ASR and MT data to explore the potential of our model. As shown in Table 5, U-XLM achieves obvious improvement in S2ST from the multi-task learning under the expanded data setting. Furthermore, the performance across different tasks (such as ASR, ST, MT, and TTS) verifies the capability of the general modeling in the decoder-only architecture. While the traditional paradigm requires a complex approach to integrate multi-task learning, the language modeling framework offers an alternative by simply modifying the prompt to construct the training data.

## 5.3 ZERO-SHOT CROSS-LINGUAL UNIT-TO-SPEECH

We select samples with a duration between 4 and 10 seconds from LibriSpeech (Panayotov et al., 2015) `dev-clean` set to evaluate the zero-shot cross-lingual unit-to-speech module. And we randomly choose one audio from EMIME as the Chinese speech prompt.

Table 6 shows the resynthesis performance of different speech synthesizers. Our TTS obtains better performance in both ASV and naturalness. We attribute the increase of WER to the difference in the amount of semantic information carried by phonemes and unsupervised units. This is consistent with the observation reported in the work of mHuBERT and AudioLM. If we remove the duration model from the U2S, the WER increases dramatically. Our guess is that the unit itself does not contain as much duration information as the phonemes. Therefore the duration model is essential when using unsupervised units.

We further train our own multilingual HuBERT model (mHuBERT_zh_en) with a combination of Chinese and English data. The model size is the same as the HuBERT-large model in (Hsu et al., 2021). We have observed a substantial reduction in the WER metric when utilizing the semantic units generated from mHuBERT_zh_en. Thus, we believe that a multilingual universal representation model trained with more parameters and data can generate better semantic units. We do not use mHuBERT_zh_en in our S2ST experiment because we need the mHuBERT (Lee et al., 2022c) to run the English→Spanish experiment. The benefit of using mHuBERT_zh_en to the overall S2ST is left for future work.

## 6 CONCLUSION AND FUTURE WORK

This paper presents a new framework for speech-to-speech translation (S2ST) based on semantic units. The framework consists of three LMs: a translation LM, a duration LM and a speech synthesis LM. Through comprehensive experimentation, we provide evidence that our unit-based S2ST system surpasses existing systems in terms of ASR-BLEU, ASV, and naturalness metrics. Importantly, our system demonstrates its effectiveness in scenarios involving unwritten languages, where there is a lack of Spanish text transcripts for reference.

Given the significant impact of semantic unit quality on our system's performance, future research will focus on improving the generation of a more refined set of discrete units. We aim to explore techniques and methodologies that can contribute to enhancing the quality and diversity of the generated semantic units. Additionally, we plan to investigate how the system's performance can be further enhanced by scaling up parameters and expanding the available training data. By exploring the effects of increased model capacity and larger datasets, we anticipate uncovering potential improvements in system accuracy and overall effectiveness.

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

# A APPENDIX

## A.1 DATASETS

We use a set of several datasets to train U-XLM model: GigaS2S, WenetS2S, LibriLight and some in-house ASR, MT datasets. Table 7 shows the detailed descriptions and statistics.

| Type | Dataset | Language | Size | Domain |
|------|---------|----------|------|--------|
| ASR | LibriLight | En | 60K hours | audiobook |
| | In-house | Zh | 60K hours | - |
| MT | In-house | Zh ↔ En | 44M sents | - |
| S2S | GigaS2S | En → Zh | 10K hours | audiobook, podcasts, youtube |
| | WenetS2S | Zh → En | 10K hours | youtube |

Table 7: Training data of U-XLM model. "-" means the dataset doesn't belong to some specific domains.

## A.2 MODEL SETTINGS

### A.2.1 S2UT

In the S2UT front-end, U-XLM's model architecture is a unidirectional Transformer decoder consisting of 48 layers with hidden size 1600, feed-forward network (FFN) size 6400, and 25 attention heads. The total parameters are 1.6B. U-XLM is trained on 8/32 NVIDIA TESLA A100 80GB GPUs with a batch size of 3072 tokens per GPU for 500k steps.

### A.2.2 U2S

In the U2S back-end, the U-SLM consists of 12 transformer layers. Each of these layers comprises 16 attention heads, an attention dimension of 1024, and an FFN dimension of 4096 in both the autoregressive (AR) model and non-autoregressive (NAR) model. We train the models using 8 NVIDIA TESLA A100 80GB GPUs, with a batch size of 8 utterances per GPU for 800k steps. Training for all steps takes about 5 days.

