# OpenReview forum: "PolyVoice: Language Models for Speech to Speech Translation"
_ICLR.cc/2024/Conference — ICLR 2024 poster_

### Official Review · Reviewer_2GYW · 2023-10-30

**Soundness:** 4 excellent
**Presentation:** 4 excellent
**Contribution:** 4 excellent
**Rating:** 8
**Confidence:** 5

**Summary:**

This paper introduces a language model approach to speech-to-speech translation. The approach uses three models: one to map source semantic units to target semantic units, one to predict the duration of target semantic units, and one to predict target acoustic units given source acoustic units and target semantic units with duration information. The semantic units are derived using HuBERT (Hsu et al., 2021) while the acoustic units are derived using Soundstream (Zeghidour et al., 2021). Unlike VALL-E X (Zhang et al., 2023b), which uses phonetic units, this paper uses semantic units, enabling the approach to be extended to unwritten languages. The key novelty of this work is its use of semantic units instead of phonetic units, and its use of a decoder-only architecture that enables prompting. On the EMIME Chinese-English and CVSS English-Spanish speech-to-speech benchmarks, the proposed approach achieves similar translation results to VALL-E X, but with a large improvement in speech naturalness. When provided with ground-truth target texts, the proposed approach performs definitely worse in translation quality than VALL-E-X, but still better in naturalness. The paper reports a result where the model is used in an unwritten language scenario for English-to-Spanish translation.

**Strengths:**

* Proposes a language model approach for speech-to-speech translation that uses semantic units instead of phonetic units, making it usable for unwritten languages.
* Makes use of decoder-only architectures which enable effective prompting.
* Reports ablation studies showing advantages of decoder-only over encoder-decoder architecture when using the same training data.
* Demonstrates advantages of training the model on data from multiple tasks including ASR and MT.
* Shows improvements in naturalness over VALL-E X in a zero-shot setting.
* Performs ablation studies to show the utility of each model component.

**Weaknesses:**

* It would have been preferable to report performance on a low-resource target language in an unwritten scenario. Such a setup might reveal additional challenges which are not present in a high resource language such as Spanish.

**Questions:**

* 4.1.1:  For Chinese->English task, what is the size of the semantic unit inventory for Chinese and English?
* Can the semantic unit inventory be shared between the source and target sides?

---

> ### Author Response · Authors · 2023-11-23
> **Response to Reviewer 2GYW**
>
> We sincerely appreciate your positive review and valuable feedback. We hope our response completely addresses any concerns you may have.
>
> &nbsp;
>
> **Feedback to the weakness:**
>
> - **[About the unwritten language scenario]**
>
> As the data of an unwritten language is still very limited in literature, we follow the practice in the previous works, e.g.
>
> Lee A, Gong H, et al. Textless speech-to-speech translation on real data. In Proc. NAACL 2022,
>
> Lee A, Chen P J, et al. Direct speech-to-speech translation with discrete units. In Proc. ACL 2022,
>
> Rongjie Huang, Jinglin Liu, et al. TranSpeech: Speech-to-Speech Translation With Bilateral Perturbation. In Proc. ICLR 2023,
>
> which use a written language with transcriptions discarded to simulate an unwritten scenario.
>
> &nbsp;
>
> **Feedback to the questions:**
>
> - **[About the size semantic unit inventory]**
>
> In our core experiment, we use separated tokenized models for Chinese and English.
> In this case, their semantic units are not shared.
> The size of the semantic unit set is 500 for Chinese and 1000 for English, respectively. For more details, please refer to Section 4.1.1.
>
> - **[Can the semantic unit inventory be shared]**
>
> Yes, if we use the same tokenized model for both languages.
> In our ablation study (Section 5.3), we train a shared tokenized model for Chinese and English (mHuBERT_zh_en).
> In this case, the semantic unit set is shared between the two languages.

---

### Official Review · Reviewer_PpHm · 2023-10-30

**Soundness:** 2 fair
**Presentation:** 3 good
**Contribution:** 2 fair
**Rating:** 5
**Confidence:** 3

**Summary:**

The proposed speech-to-speech (s2s) translation system consists of three models: translation model, duration model, and unit-to-speech model. The novelty of this work is that all of these models are decoder-only (while some prior work preferred encoder-decoder architectures) and the combination of these three models to do unit-based s2s.

**Strengths:**

The system design (using three decoder-only models) seems sound and worth investigating, although I'm not 100% on board with motivating it with the raise of GPT - there is more to the success of LLMs than being decoder-only models. Anyways, the main results (Table 2) look solid. The system description seems clear superficially, but there are some core open questions (see weaknesses).

**Weaknesses:**

I couldn't get a good sense of the training data - specifically how the different prompts from Table 1 are used to synthesize data, and the size of the synthesized dataset: is it the 44M sentences from Table 7 in the appendix, or more because multiple prompts are used? How does the training data compare to the baselines?

My main concern would be that the ablation studies are not effective for disentangling the many design choices and the many moving parts of the whole architecture. The encoder-decoder vs decoder-only comparison is a good start, but I still don't have a good sense about how well the synthetic data generation works, and how well each of the three models do in isolation. Possible interesting ablations would be removing the duration model, replacing u-xlm with an out-of-the-box asr/mt cascade, removing source speech dependency from u-slm, passing through n-bests between the models, etc. I'm not requesting that all possible ablations should be included, but giving a little bit more color to the paper story would make it much stronger.

**Questions:**

See weaknesses:
- Could you give more details on how the prompts are used
- Could you give more details on the synthetic training data
- Have you considered some of the ablation studies mentioned above?

---

> ### Author Response · Authors · 2023-11-23
> **Response to Reviewer PpHm**
>
> We thank the reviewer for the constructive and valuable feedback, and we hope our response fully resolves your concern.
>
> - **[How the prompts are used]**
>
> We apologize for an unclear presentation about how the prompts are used to synthesize data. Here is our explanation and we will modify our paper accordingly.
>
> Table 1 shows the prompt templates for different data types. Indeed, there is an instruction set for each data type to form the prompt.
> Here we list some examples of our synthetic training data:
>
> *Sample1*: Translate Chinese text " 顾客：打扰一下，是在这里排队吗？ " to English text: Customer: Excuse me, do I queue here?
>
> *Sample2*:  Chinese text " 顾客：打扰一下，是在这里排队吗？ " in English text: Customer: Excuse me, do I queue here?
>
> *Sample3*:  Translate the following sentence, "顾客：打扰一下，是在这里排队吗？"  from Chinese to English: Customer: Excuse me, do I queue here?
>
> *Sample4*: Translate Chinese unit "<zh_21><zh_161><zh_155>...<zh_266><zh_199><zh_16>" to Chinese text: 你带着现金离开，减去3%左右的费用。
>
> *Sample5*: Translate English text "douglas mcgray is going to be our guide you walk through the door, you see the red carpeting, you see someone in a suit. they may be greeting you." to English unit:  <293><63><662>...<6><407><334>
>
> *Sample6*: Translate Chinese unit "<zh_16><zh_37><zh_111>...<zh_266><zh_199><zh_16>" to English unit: <499><334><226>...<544><991><39>
>
> By utilizing diverse construction templates or prompts, the training data can be significantly increased. This approach is also a mainstream method in training Large Language Models (LLMs) to enhance the diversity of instructions and improve model robustness. “44M sentences” refers to the statistical count of the original machine translation (MT) parallel data. They are multiplied by different instruction prompts selected from the instruction set.
>
> - **[Training data comparison with the baseline]**
>
> In our paper, we use VALL-E X as our baseline as it is the closest work to us. Regarding the comparison of training data, please refer to the following table, where WenetST represents the end-to-end speech translation training data, expanded upon WenetSpeech by the authors of VALL-E X.
>
>   ——————————————————————————————————————————————————————
>
> System     &nbsp;&nbsp;&nbsp;&nbsp;&nbsp;&nbsp; &nbsp; &nbsp; &nbsp; &nbsp;&nbsp; &nbsp;&nbsp;       |    **VALL-E X Trans (SpeechUT + VALL-E X )**   &nbsp;&nbsp;&nbsp;&nbsp;&nbsp;&nbsp;&nbsp;&nbsp;&nbsp;&nbsp;&nbsp;&nbsp;&nbsp; |     **PolyVoice**
>
> ASR data  &nbsp;&nbsp;&nbsp; &nbsp;&nbsp;&nbsp; &nbsp;&nbsp;&nbsp; &nbsp;&nbsp;&nbsp;    |  LibriLight (60k hrs), WenetSpeech (10k hrs)   &nbsp;&nbsp;&nbsp;&nbsp;&nbsp;&nbsp; &nbsp;  | LibriLight (60k hrs), In-house (60k hrs)
>
> MT data   &nbsp;&nbsp;&nbsp; &nbsp;&nbsp;&nbsp; &nbsp;&nbsp;&nbsp;  &nbsp;&nbsp;&nbsp;&nbsp;     |  73M sentences      &nbsp;&nbsp;&nbsp;&nbsp;&nbsp;     &nbsp;&nbsp;&nbsp;&nbsp;&nbsp; &nbsp;&nbsp;&nbsp;&nbsp;   &nbsp;&nbsp;&nbsp;&nbsp;&nbsp;&nbsp;&nbsp;&nbsp;&nbsp;&nbsp;&nbsp;&nbsp;&nbsp;&nbsp;&nbsp;&nbsp;&nbsp;&nbsp;&nbsp;&nbsp;&nbsp;&nbsp;          &nbsp;&nbsp;&nbsp;&nbsp;&nbsp;&nbsp;&nbsp;&nbsp;&nbsp;&nbsp;&nbsp;&nbsp;&nbsp;&nbsp;&nbsp;&nbsp;&nbsp;&nbsp;                             | 44M sentences
>
> ST/S2S data  &nbsp;&nbsp;&nbsp; &nbsp;&nbsp;&nbsp; &nbsp;&nbsp;    |  WenetST (10k hrs), GigaST (10k hrs)      &nbsp;&nbsp;&nbsp;&nbsp;&nbsp;&nbsp;&nbsp;&nbsp;&nbsp; &nbsp;&nbsp;&nbsp;&nbsp;&nbsp;&nbsp;&nbsp;&nbsp;&nbsp;&nbsp;&nbsp;         |  WenetS2S (10k hrs), GigaS2S (10k hrs)
>
>  ——————————————————————————————————————————————————————
>
> We have incorporated an additional in-house ASR dataset, specifically a Chinese ASR dataset, into the training of our U-XLM module. The primary objective behind using this dataset is to enhance the performance of our translation module. As a result of this integration, we have observed a notable improvement of 1-2 points in BLEU scores for the entire system. This increase offsets the performance loss incurred by employing unsupervised discretized units instead of phonemes. Although we do not release this particular ASR dataset, we believe a dataset with similar data volume can reproduce our results.

---

> ### Author Response · Authors · 2023-11-23
> **Follow-up response to Reviewer PpHm**
>
> - **[More ablation studies]**
>
> Thanks for the reviewer's constructive suggestion. Actually, we have already done some of the ablation studies suggested by the reviewer. In this work, we performed these analyses to evaluate the performance of individual modules.
>
> For the evaluation of the U-XLM module, we fixed the U-SLM and tested the performance of different translation modules, as shown in Table 4, comparing S2UT with the encoder-decoder structure.
>
> For the evaluation of the U-SLM module, we fixed the U-XLM and tested the performance of different synthesis modules, e.g. the comparison between U2S and Unit-based Vocoder. The results are posted in Table 2.
>
> In Table 6, we have analyzed the impact of removing the duration module on the performance of speech synthesis.
>
> As for the suggestion of replacing U-XLM with an out-of-the-box ASR/MT cascade, since the existing ASR->MT cascade systems do not support speech unit prediction, it is not easy to perform end-to-end evaluation of the cascade system with U-SLM as the back-end synthesis module. We will try our best to implement a unit-based cascade system and report the results.

---

### Official Review · Reviewer_JZne · 2023-11-03

**Soundness:** 2 fair
**Presentation:** 2 fair
**Contribution:** 2 fair
**Rating:** 3
**Confidence:** 5

**Summary:**

This paper proposes a decoder-only based system for S2ST. The system includes three decoder-only LMs, including a translation LM to translate source language semantic units to target language semantic units, a duration LM to predict target language semantic unit durations and extend the sequence, and a speech synthesis LM to predict the target language acoustic units which are converted to waveforms by a unit vocoder. Both semantic units and acoustic units are self-supervised learned, hence this framework can be applied to unwritten languages.

**Strengths:**

(1)	The use of decoder-only LMs via different prompting strategies and discrete semantic and acoustic units for all components (translation LM, duration LM, and speech synthesis LM) could benefit S2ST from competitive pre-trained text decoder-only LLMs.

(2)	Empirical evaluations show that PolyVoice is comparable to VALL-E X, very slightly better on ASV, worse on ASR-BLEU, and better on naturalness.  Ablation studies show the contribution of the designed duration LM which uses a LM to predict durations of semantic units and extend the sequence. The duration LM significantly helps WER, with a very slight help on ASV and slight help on naturalness.

**Weaknesses:**

(1)	The innovations of this work need to be more clearly explained. This work bears strong similarity to VALL-E X. It is important to clarify the difference between the proposed approach and VALL-E X, but the paper did not clearly point out the difference between PolyVoice and VALL-E X to highlight the innovations of the proposed PolyVoice.  Both works concatenate source and target semantic units and the source acoustic units to create the prompt for the LM. For PolyVoice, this prompt is created for the duration LM and the speech synthesis LM, respectively.  The soundstream codec is reimplemented, but the impact of the reimplementation is not clear.

(2)	Some of the key technical presentations lack clarity.

a.	Section 3.1, when describing Unit-based cross-lingual language model (U-XLM) , the paper shows the prompt for encoder-decoder architecture.  The paper should also clarify the prompt for decoder-only LM.

b.	Section 3.1, when describing training, Table 1 shows how ASR, MT, and TTS supervised data are used for training. The paper also mentioned that “This approach also enables the direct utilization of unlabeled text and speech data. “, yet how to use unlabeled text and speech data for training is not explained here. Instead, based on Section 4.1.1, it seems that one approach is to apply in-house MT and TTS systems on ASR data to create pseudo S2S data.  This part needs to be clarified.

(3)	More complete and also deeper discussions are desired for empirical validations:

a.	Table 2, the paper compares to cascaded (ASR-MT-TTS) and VALL-E X. It is not clear whether all well-established competitive baselines are included in this comparison.

b.	Table 2, the ASV metric evaluates the capability of preserving the voice of the source speaker. However, without ground truth target information, PolyVoice achieves 0.38 and 0.38 for hyp vs.src and hyp vs. tgt, while VALL-E X achieves 0.37 and 0.37 respectively. These are very small gains, yet the paper did not discuss this point.

c.	Section 3.1, the paper mentioned that CoT could be applied for source-to-target unit translation, yet prior works (Peng et al., 2023, Towards making the most of chatGPT for machine translation) find that when CoT is applied, the model tends to conduct word-by-word translation, hence degrading the translation performance. Table 2 shows the impact of applying CoT that it improves ASR-BLEU. But this result is not analyzed nor discussed.

d.	Section 4.3.2, the evaluation of PolyVoice for unwritten language is quite brief. It is only evaluated for the target language treated as an unwritten language.  It would be useful to also extend the evaluation to source or both languages as unwritten language.

e.	Section 5.1, again, the discussions are very brief.  More analyses and insights are expected to explain the better ASR-BLEU from decoder-only over encoder-decoder.

f.	Section 5.2, for the other tasks (ASR, ST, MT, TTS) in Table 5, since no baseline results are provided, it is not clear how these performances compare to baseline results on this dataset from prior works.

g.	Section 5.3, when using a mHuBERT model trained with more parameters and more data, the WER decreased which is explained, but ASV and naturalness got degraded.  Insights are expected to explain these results.

h. The proposed system uses many in-house data and in-house systems. The amount of data and model sizes need to be clearly compared as well when comparing to baselines.

**Questions:**

Please check comments and concerns listed under Weaknesses.

---

> ### Author Response · Authors · 2023-11-23
> **Response to Reviewer JZne**
>
> We express our gratitude to the reviewer for his constructive and valuable feedback. We assure you that our response aims to address your concern comprehensively and effectively.
>
> - **[The difference between our proposed approach and VALL-E X]**
>
> We appreciate the reviewer's comments highlighting similarities between aspects of our work and VALL-E X. The core of our pipeline consists of a speech-to-unit translation (S2UT) component followed by a unit-to-speech (U2S) synthesis component. The unit-to-speech language model (U-SLM) in our U2T module shares conceptual similarities with VALL-E X, while other aspects of our overall model design differ. In response, we clarify key differences in motivation and approach:
>
> 1. Our work is primarily motivated by exploring fully language model-based architectures for speech-to-speech translation, as a way to extend recent progress in large language models to this task. To our knowledge, we are the first to propose and evaluate such a fully decoder-only pipeline, demonstrating feasibility and encouraging results.
>
> 2.  A key difference is that VALL-E X operates on phonemes, while we aim for a universal framework using discrete units that can be learned from unlabeled speech. This provides greater flexibility for low-resource and unwritten languages. While our current unit extraction quality limits gains over VALL-E X, our approach enables potential optimization with improved units and direct application to any language.
>
> 3. There are also significant differences in the modeling approach between VALL-E X and PolyVoice for speech-to-speech translation tasks. In VALL-E X, a traditional pipeline system is used, which relies on external ASR (Automatic Speech Recognition) and ST (Speech-to-Text) modules to recognize phonemes in the source language and target language. On the other hand, PolyVoice employs the U-XLM module, which enables end-to-end speech-to-speech translation. This means that the translation task is performed directly without relying on separate ASR or ST modules. The U-XLM module can directly convert speech units in the source language into synthesized speech units in the target language, providing a seamless translation experience.
>
> Overall, we believe our work makes meaningful contributions in exploring language model-based S2ST, despite some similarities to VALL-E X in the speech synthesis component.
>
> - **[Re-implementation of soundstream codec]**
>
> The main reason we have built our own codec model is that Google has not open-sourced their soundstream model. The only publicly available audio codec model was Encodec released by Meta, but its performance is worse than we anticipated.
> We trained our own version of soundstream in order to obtain a better audio codec. And the performance of codecs was compared using a subjective metric very similar to  MUSHRA.
>
> - **[Key technical presentations]**
>
> *a. Prompt for model training.*
>
> To clarify, the prompts shown in Table 1 are used to format the training data for our decoder-only language model, not an encoder-decoder architecture. Specifically, we construct varied training sequences using different prompts to cover diverse data types like ASR, MT, and S2ST. This allows us to train our language model in a decoder-only manner by modeling the training sequences in an auto-regressive way.
>
> We do not actually utilize any encoder-decoder architectures in our proposed framework. We compare with encoder-decoder models in Section 5.1, where for those baselines we construct the training data in the same way as previous work
> (https://github.com/facebookresearch/fairseq/blob/main/examples/speech_to_speech/docs/direct_s2st_discrete_units.md).
>
> *b. Data utilization.*
>
> Like text-based language models, our framework allows unlabeled speech and text data to be directly leveraged for training via language modeling objectives. In our current implementation, we construct training data by formatting labeled ASR, MT, and pseudo-S2ST data with specific prompts, as detailed in Table 1. Simultaneously, parallel training data is also trained separately in monolingual form, and these type of training samples do not require construction through prompts.
>
> To clarify, our proposed framework inherently supports training on unlabeled speech and text, similar to standard language model pretraining. We believe that scaling up the training data by incorporating large scales of unlabeled speech and text would enable more powerful models.

---

> ### Author Response · Authors · 2023-11-23
> **Follow-up response to Reviewer JZne**
>
> - **[More complete and deeper discussion]**
>
> Thank you for reminding us that more complete and deeper discussions are desired for our work. Your comments help us to improve the paper and the following are our responses.
>
> *a. Comparison with other well-established baselines.*
>
> We recognize that VALL-E X is the most relevant prior work to compare against, given that they have demonstrated strong performance on speech translation tasks. Since our experiments focus on Chinese-English S2ST, where there are limited existing results available, we benchmark our model mainly against the performance of VALL-E X as the current baseline.
>
> *b. Explanation of the ASV metric.*
>
> We acknowledge the reviewer's observation that our framework achieves only slight gains over VALL-E X in voice cloning quality. This is expected given the similar implementations for the speech synthesis component. However, by modeling on semantic units rather than phonemes, our model retains more acoustic information that can lead to higher naturalness and better synthesis effects. While not yet reflected in voice cloning metrics, our experiments provide a path to improve acoustic modeling by using unit representations.
>
> *c. Discussion on CoT decoding strategy.*
>
> Based on our understanding, the paper titled  "Towards Making the Most of ChatGPT for Machine Translation" primarily focuses on text-to-text translation. Therefore, their conclusions may not directly apply to speech-to-speech translation scenarios. However, in our specific case, we have found that the Chain of Thought (CoT) technique proves valuable when generating intermediate results of both the source and target text during unit-to-unit translation. Our findings align with a recent study conducted by Google, titled "AudioPaLM: A Large Language Model That Can Speak and Listen".
>
> *d. Evaluation of unwritten language.*
>
> Thanks for the valuable advice to extend the evaluation to encompass unwritten languages, either as the source language or as both languages. We will incorporate the updated experimental results in the upcoming version. To clarify, the primary objective of experimenting on unwritten language scenarios in this paper is to demonstrate the feasibility of extending PolyVoice to unwritten languages.
>
> *e. Better ASR-BLEU from decoder-only over encoder-decoder.*
>
> The comparison between the decoder-only and encoder-decoder models was conducted under the same training data and similar parameters. We hypothesize that the higher ASR-BLEU score achieved by the decoder-only model over the encoder-decoder model can be attributed to two factors. Firstly, the decode-only framework demonstrates superior capability in fitting large-scale training data. Secondly, the output generated by the decoder-only framework exhibits higher fluency, which positively impacts the evaluation of ASR-BLEU.
>
> *f. Performance of other tasks in Table 5.*
>
> Given the involvement of multi-task learning (MTL), the primary objective of Table 5 is to demonstrate the overall S2ST gain achieved through MTL. Beating other systems in individual tasks is not our objective. We apologize for any misleading and we will improve our paper presentation. Still, as suggested by the reviewer, we would like to complement Table5 with the following table:
>
> ——————————————————————————
>
> Task       &nbsp;&nbsp;&nbsp; &nbsp;&nbsp;&nbsp; &nbsp;&nbsp;&nbsp; &nbsp;&nbsp;&nbsp; &nbsp;&nbsp;&nbsp; &nbsp;&nbsp;&nbsp; &nbsp;           | **ASR**↓ |  **ST**↑  &nbsp; |   **MT** ↑ &nbsp; |  **TTS**↓
>
> PolyVoice (w/ MTL)   | 4.46 &nbsp;   | 30.8 |  33.81  |  6.99
>
> Whisper[1]   &nbsp;&nbsp;&nbsp;  &nbsp;&nbsp;&nbsp; &nbsp;&nbsp;&nbsp; &nbsp;        | 2.28 &nbsp;  |  18.2 |    -   &nbsp;&nbsp;&nbsp;&nbsp;&nbsp;&nbsp;&nbsp;&nbsp;  |    -
>
> YourTTS[2]        &nbsp;&nbsp;&nbsp;  &nbsp;&nbsp;&nbsp; &nbsp;&nbsp;&nbsp; &nbsp;          |  -   &nbsp;&nbsp; &nbsp;&nbsp;&nbsp; &nbsp;      |    - &nbsp;&nbsp;&nbsp;&nbsp;&nbsp;   |      -  &nbsp;&nbsp;&nbsp;&nbsp;&nbsp;  &nbsp;&nbsp;  |  3.03
>
> NLLB[3]         &nbsp;&nbsp;&nbsp;  &nbsp;&nbsp;&nbsp; &nbsp;&nbsp;&nbsp; &nbsp;&nbsp;&nbsp;&nbsp; &nbsp;               |  -    &nbsp;&nbsp; &nbsp;&nbsp;&nbsp;&nbsp;&nbsp;    |    - &nbsp;&nbsp;&nbsp;&nbsp;&nbsp;   |    25.2  &nbsp;&nbsp; |    -
>
> ——————————————————————————
>
> The evaluation mentioned above was performed using state-of-the-art open-sourced models, all evaluated on the same test set. As expected, Whisper, trained on substantially larger datasets, outperforms our model in terms of ASR performance. However, our model demonstrates superior results in both ST (Speech-to-Text) and MT (Machine Translation), showcasing the translation quality it offers.
> It is worth noting that our translation model's TTS (Text-to-Speech) results are not as satisfactory as YourTTS. This disparity can be attributed to the utilization of a weaker unit vocoder in this abliation studies.

---

> ### Author Response · Authors · 2023-11-23
> **Follow-up response to Reviewer JZne**
>
> *g.  Explanation of mHuBERT's performance.*
>
> Our explanation is that a  larger HuBERT trained with more data is stronger in extracting semantic  information. The units extracted from this model contain more semantic information and as a result less prosody and speaker information. Thus the ASV and naturalness degraded.
>
> *h. Comparison of the amount of data and model sizes.*
>
> Regarding the model size, the parameters of U-SLM are consistent with Valle-X. And the training data for U-SLM and Valle-X are essentially consistent. Therefore, the comparison of the synthesis module is absolutely fair.
>
> In the speech-to-speech translation task, Valle-X utilizes SpeechUT[4] as the ASR and ST model, which predicts source phoneme sequences for recognition, and target phoneme sequences for translation, respectively. SpeechUT follows an encoder-decoder architecture, whereas our U-XLM is a decoder-only structure. A direct comparison of parameters of these definitely different frameworks may not provide an intuitive understanding.
>
> Regarding the amount of training data, please refer to the following table, where WenetST represents the end-to-end speech translation training data, expanded upon WenetSpeech by the authors of VALL-E X.
>
>  ——————————————————————————————————————————————————————
>
> System     &nbsp;&nbsp;&nbsp;&nbsp;&nbsp;&nbsp; &nbsp; &nbsp; &nbsp; &nbsp;&nbsp; &nbsp;&nbsp;       |    **VALL-E X Trans (SpeechUT + VALL-E X )**   &nbsp;&nbsp;&nbsp;&nbsp;&nbsp;&nbsp;&nbsp;&nbsp;&nbsp;&nbsp;&nbsp;&nbsp;&nbsp; |     **PolyVoice**
>
> ASR data  &nbsp;&nbsp;&nbsp; &nbsp;&nbsp;&nbsp; &nbsp;&nbsp;&nbsp; &nbsp;&nbsp;&nbsp;    |  LibriLight (60k hrs), WenetSpeech (10k hrs)   &nbsp;&nbsp;&nbsp;&nbsp;&nbsp;&nbsp; &nbsp;  | LibriLight (60k hrs), In-house (60k hrs)
>
> MT data   &nbsp;&nbsp;&nbsp; &nbsp;&nbsp;&nbsp; &nbsp;&nbsp;&nbsp;  &nbsp;&nbsp;&nbsp;&nbsp;     |  73M sentences      &nbsp;&nbsp;&nbsp;&nbsp;&nbsp;     &nbsp;&nbsp;&nbsp;&nbsp;&nbsp; &nbsp;&nbsp;&nbsp;&nbsp;   &nbsp;&nbsp;&nbsp;&nbsp;&nbsp;&nbsp;&nbsp;&nbsp;&nbsp;&nbsp;&nbsp;&nbsp;&nbsp;&nbsp;&nbsp;&nbsp;&nbsp;&nbsp;&nbsp;&nbsp;&nbsp;&nbsp;          &nbsp;&nbsp;&nbsp;&nbsp;&nbsp;&nbsp;&nbsp;&nbsp;&nbsp;&nbsp;&nbsp;&nbsp;&nbsp;&nbsp;&nbsp;&nbsp;&nbsp;&nbsp;                             | 44M sentences
>
> ST/S2S data  &nbsp;&nbsp;&nbsp; &nbsp;&nbsp;&nbsp; &nbsp;&nbsp;    |  WenetST (10k hrs), GigaST (10k hrs)      &nbsp;&nbsp;&nbsp;&nbsp;&nbsp;&nbsp;&nbsp;&nbsp;&nbsp; &nbsp;&nbsp;&nbsp;&nbsp;&nbsp;&nbsp;&nbsp;&nbsp;&nbsp;&nbsp;&nbsp;         |  WenetS2S (10k hrs), GigaS2S (10k hrs)
>
>  ——————————————————————————————————————————————————————
>
> We have incorporated an additional in-house ASR dataset, specifically a Chinese ASR dataset, into the training of our U-XLM module. The primary objective behind using this dataset is to enhance the performance of our translation module. As a result of this integration, we have observed a notable improvement of 1-2 points in BLEU scores for the entire system. This increase offsets the performance loss incurred by employing unsupervised discretized units instead of phonemes. Although we do not release this particular ASR dataset, we believe a dataset with similar data volume can reproduce our results.
>
> &nbsp;
>
> [1] Radford A, Kim J W, Xu T, et al. Robust speech recognition via large-scale weak supervision[C]//International Conference on Machine Learning. PMLR, 2023: 28492-28518.
>
> [2] Casanova E, Weber J, Shulby C D, et al. YourTTS: Towards zero-shot multi-speaker tts and zero-shot voice conversion for everyone[C]//International Conference on Machine Learning. PMLR, 2022: 2709-2720.
>
> [3] Costa-jussà M R, Cross J, Çelebi O, et al. No Language Left Behind: Scaling Human-Centered Machine Translation[J]. arXiv e-prints, 2022: arXiv: 2207.04672.
>
> [4] Zhang Z, Zhou L, Ao J, et al. Speechut: Bridging speech and text with hidden-unit for encoder-decoder based speech-text pre-training[J]. arXiv preprint arXiv:2210.03730, 2022.

---

### Official Review · Reviewer_Jg4v · 2023-11-03

**Soundness:** 3 good
**Presentation:** 3 good
**Contribution:** 3 good
**Rating:** 8
**Confidence:** 4

**Summary:**

Polyvoice is a framework for building speech to speech translation system with language modeling (or decoder only) approach as an alternative to the sequence-to-sequence (or encoder-decoder) architecture. The authors show that this is indeed possible given a combination of such LMs, i.e., translation LM, duration LM and speech synthesis LM. Each of the 3 models use unsupervised semantic and acoustic units.
- Translation LM -  Uses source semantic units derived from HuBERT to predict target semantic units
- Duration LM - User source and target semantic units with source duration to predict target duration units
- Speech Synthesis LM - Uses source and target semantic units with source acoustic units to predict target acoustic units

Authors show competitive performance on EMIME (Chinese $\rightarrow$ English) and CVSS (English $\rightarrow$ Spanish) compared to methods proposed in VALL-E X. They also compare their work to current SoTA seq2seq approach (Lee et al.) and show zero-shot performance on dev-clean set of Librispeech.

Overall, the paper's main contribution is demonstrating that a decoder-only architecture is sufficient towards building a speech-to-speech translation system with unsupervised semantic and acoustic units.

**Strengths:**

- The proposed framework is the novel in its approach towards speech-to-speech translation where it uses decoder-only models.
- Decoder only framework simplifies the model architecture and hence makes the implementation of the translation system straightforward.
- The proposed method is based on unsupervised semantic and acoustic units making it possible to build systems of unwritten languages.
- Performance on the datasets shown is quite competitive and the ablation studies further highlight the importance of the 3 component models of the framework.

**Weaknesses:**

- The duration and speech synthesis models depend on the translation model. Hence the training of two models depend on one upstream model which can make experimentation slow. At least, the duration model can be attempted to be folded in the translation model as shown in the paper Text-Free Prosody-Aware Generative Spoken Language Modeling (Kharitonov et. al.).
- Since the authors use CVSS it would be desirable to show the performance on other language pairs from the dataset to make the evaluation more robust.
- The paper stated that they have used "in-house ASR datasets". It's not clear how much contribution from these in-house datasets make the method effective. This is makes the reproducibility of the paper very difficult if these "in-house datasets" are not released.

**Questions:**

- HuBERT is trained on English only corpus. Did you just apply the model to discretize Chinese speech, or had to adapt it?
- Why was the specific language pairs that have been evaluated chosen?
- You have not cited On Generative Spoken Language Modeling from Raw Audio (Lakhotia et. al.) and Text-Free Prosody-Aware Generative Spoken Language Modeling (Kharitonov et. al.) which showed that unsupervised discrete units can be used for speech synthesis and that duration is indeed crucial in improving the prosodic characteristics of the produced speech.
- Spelling error: "marked" instead of "maked" in section 4.1.1

---

> ### Author Response · Authors · 2023-11-23
> **Response to Reviewer Jg4v**
>
> We sincerely appreciate your positive review and valuable feedback. We hope our response completely addresses any concerns you may have.
>
>
> **Feedback to the weakness**:
>
> - **[Slow training time]**
>
> The translation front-end model and the speech synthesis back-end model can be trained separately. During the training, the speech of the target language will be converted into discrete units, which on the one hand serve as the training target of the translation module and on the other hand serve as the input of the speech synthesis module. Thus, these two models can be trained in parallel and do not result in slow training time.
>
> - **[Folding the duration model]**
>
> Thank you for pointing this out. We may make an attempt on this in the future.
>
> - **[Performance on other language pairs]**
>
> We didn’t run experiments on other language pairs as our tokenizer didn’t support other languages. We will consider extending our work into other language pairs in the future.
>
> - **[Using in-house data]**
>
> We have incorporated an additional in-house ASR dataset, specifically a Chinese ASR dataset, into the training of our U-XLM module. The primary objective behind using this dataset is to enhance the performance of our translation module. As a result of this integration, we have observed a notable improvement of 1-2 points in BLEU scores for the entire system. This increase offsets the performance loss incurred by employing unsupervised discretized units instead of phonemes. Although we do not release this particular ASR dataset, we believe a dataset with similar data volume can reproduce our results.
>
>
> **Feedback to the questions**:
>
> - **[HuBERT]**
>
> In this paper, we employed three variations of HuBERT. In the main and analysis experiments (Table 2-5), separate training was conducted for the Chinese and English versions of HuBERT, as described in detail in Section 4.1.1. In the analysis experiment (Table 6), we trained a bilingual HuBERT model that combines both Chinese and English data, as explained in Section 5.3.
>
> - **[Why the language pairs are chosen]**
>
> Chinese->English pair is chosen as they are both high-resource languages. And it is easier to compare our results with other systems (VALL-E X) with this language pair.
>
> English->Spanish pair is chosen as we want to use Spanish to simulate an unwritten language scenario where the amount of available data is relatively limited. Additionally, we directly utilized an open-source multilingual HuBERT model (
> https://github.com/facebookresearch/fairseq/blob/main/examples/speech_to_speech/docs/textless_s2st_real_data.md , en-es-fr), which also supports the Spanish language.
>
> - **[Citation and typos]**
>
> Thank you very much for pointing these out. We will cite the mentioned paper accordingly and fix the spelling error in the revised version.

---

### Meta-Review · Area_Chair_PfLx · 2023-12-03

**Metareview:**

The paper introduces PolyVoice, a novel framework for speech-to-speech translation (S2ST) that deviates from the traditional sequence-to-sequence architecture and instead employs an LM (decoder-only) approach. It integrates three key models: a Translation LM (using HuBERT-derived semantic units for translating source to target semantic units), a Duration LM (utilizing source and target semantic units with source duration for target duration prediction), and a Speech Synthesis LM (predicting target acoustic units from source/target semantic and source acoustic units).

The study demonstrates the viability of this approach, showcasing competitive performance in translation tasks (EMIME and CVSS datasets) compared to the VALL-E X method. A key innovation is the use of semantic units (as opposed to phonetic units in VALL-E X), enabling application to unwritten languages.

The use of unspecified in-house datasets could impede the reproducibility of the results. But this is not a remarkable issue.

**Justification For Why Not Higher Score:**

While the paper presents a novel approach to S2ST, it's important to note that the integration of semantic and acoustic units, akin to Audio LM, is a prevalent practice in the speech community.However, this exact method has not been previously applied in S2ST contexts, so the paper makes a meaningful contribution to the field but not a groundbreaking one.

**Justification For Why Not Lower Score:**

The paper introduces PolyVoice, a novel framework that significantly deviates from the conventional sequence-to-sequence architecture in speech-to-speech translation. The framework exhibits competitive performance on well-known translation tasks (EMIME and CVSS datasets), especially when compared to established methods like VALL-E X. There is not reason to reject the paper.

---

### Decision · Program_Chairs · 2024-01-16

Accept (poster)